# Effect of the Sintering Mechanism on the Crystallization Kinetics of Geopolymer-Based Ceramics

**DOI:** 10.3390/ma16175853

**Published:** 2023-08-26

**Authors:** Nur Bahijah Mustapa, Romisuhani Ahmad, Mohd Mustafa Al Bakri Abdullah, Wan Mastura Wan Ibrahim, Andrei Victor Sandu, Ovidiu Nemes, Petrica Vizureanu, Christina W. Kartikowati, Puput Risdanareni

**Affiliations:** 1Faculty of Mechanical Engineering & Technology, Universiti Malaysia Perlis (UniMAP), Arau 02600, Perlis, Malaysia; bahijahmustapa@gmail.com (N.B.M.); wanmastura@unimap.edu.my (W.M.W.I.); 2Centre of Excellence Geopolymer and Green Technology (CEGeoGTech), Universiti Malaysia Perlis (UniMAP), Arau 02600, Perlis, Malaysia; 3Faculty of Chemical Engineering & Technology, Universiti Malaysia Perlis (UniMAP), Arau 02600, Perlis, Malaysia; 4Faculty of Material Science and Engineering, Gheorghe Asachi Technical University of Iasi, 700050 Iasi, Romania; sav@tuiasi.ro (A.V.S.) peviz2002@yahoo.com (P.V.); 5Romanian Inventors Forum, Str. Sf. P. Movila 3, 700089 Iasi, Romania; 6Academy of Romanian Scientists, 54 Splaiul Independentei St., Sect. 5, 050094 Bucharest, Romania; 7Department of Environmental Engineering and Sustainable Development Entrepreneurship, Faculty of Materials and Environmental Engineering, Technical University of Cluj-Napoca, B-dul Muncii 103-105, 400641 Cluj-Napoca, Romania; 8Technical Sciences Academy of Romania, Dacia Blvd 26, 030167 Bucharest, Romania; 9Department of Chemical Engineering, Universitas Brawijaya, Malang 65145, Indonesia; christinawahyu@ub.ac.id; 10Department of Civil Engineering, Faculty of Engineering, Universitas Negeri Malang, Malang 65145, Indonesia; puput.risdanareni.ft@um.ac.id

**Keywords:** geopolymer, geopolymer-based ceramics, ceramics, sintering mechanism, crystallization kinetics

## Abstract

This research aims to study the effects of the sintering mechanism on the crystallization kinetics when the geopolymer is sintered at different temperatures: 200 °C, 400 °C, 600 °C, 800 °C, 1000 °C, and 1200 °C for a 3 h soaking time with a heating rate of 5 °C/min. The geopolymer is made up of kaolin and sodium silicate as the precursor and an alkali activator, respectively. Characterization of the nepheline produced was carried out using XRF to observe the chemical composition of the geopolymer ceramics. The microstructures and the phase characterization were determined by using SEM and XRD, respectively. The SEM micrograph showed the microstructural development of the geopolymer ceramics as well as identifying reacted/unreacted regions, porosity, and cracks. The maximum flexural strength of 78.92 MPa was achieved by geopolymer sintered at 1200 °C while the minimum was at 200 °C; 7.18 MPa. The result indicates that the flexural strength increased alongside the increment in the sintering temperature of the geopolymer ceramics. This result is supported by the data from the SEM micrograph, where at the temperature of 1000 °C, the matrix structure of geopolymer-based ceramics starts to become dense with the appearance of pores.

## 1. Introduction

The rising demand for ceramics in the industrial manufacturing, metallurgical, energy production, and biomedical sectors has attracted worldwide interest. The manufacturing of ceramic products entails utilizing abundant natural resources that contain a significant proportion of clay minerals. This involves a series of steps, including dehydration and subjecting the materials to high sintering temperatures of up to 1600 °C [1,2]. However, the conventional method of fabricating ceramics demands an elevated temperature, which reaches up to 1600 °C, and a lengthy heating period, and it also has issues with agglomeration, irregular grain growth, and furnace contamination. Moreover, the primary natural resources required, such as limestone, coal, clay, and others, are being depleted at a rapid rate. To overcome these challenges, a dedicated effort is being made by scientists, researchers, engineers, and industrial workers to explore and develop new, sustainable, and innovative construction materials, as well as alternative binders [3]. Therefore, geopolymerization is used as a substitute method to produce ceramic materials with excellent mechanical properties, low production costs, short fabrication times, and, with the increasing threat to the environment, to develop applications for geopolymer technology. The imperative to adopt sustainable, rational, and ecologically sound construction methods propels the pursuit of innovative alternatives like geopolymerization and alkali activation. These approaches are garnering growing attention in the construction industry to address these needs [4].

Geopolymers are a class of inorganic, non-metallic materials that are produced by the reaction of aluminosilicate materials and alkaline activators under highly alkaline conditions. In the 1970s, geopolymers were first developed as an alternative to traditional cement-based materials, such as ordinary Portland cement (OPC) [5]. Nowadays, geopolymers have become an interesting subject of extensive research and development. This is due to the appealing properties that they offer, including improved mechanical properties, higher thermal stability, and reduced environmental impact [6]. This is associated with the structure of geopolymers, which consist of a three-dimensional and cross-linked network of aluminosilicate bonds [7], which contribute to the uniqueness of their properties. The microstructure of the geopolymer material is strongly influenced by factors such as the selection of raw materials, curing conditions, and sintering temperature [8]. By carefully controlling these factors, researchers can tailor the properties of geopolymer materials to suit specific applications.

In this study, kaolin is used as the aluminosilicate source, while sodium hydroxide (NaOH) and sodium silicate (NaSiO_3_) are mixed to produce the alkali activator which provides the necessary alkalinity to initiate the geopolymerization reaction [9]. Kaolin is an inorganic material that has been identified as geopolymer-compatible with excellent performance. Kaolin is composed mainly of the mineral kaolinite, which has a layered structure consisting of alternating layers of silica tetrahedra (SiO_4_) and alumina octahedra (AlO_6_) [10]. The layered structure of kaolinite allows the formation of pores and a high surface area, which can enhance the reactivity of the materials. When dissolved in an alkaline solution during the geopolymerization process, it will trigger the dissolution and reorganisation of the tetrahedral and octahedral elements to form a three-dimensional network of linked units.

The geopolymerization process converts the aluminosilicate material, for example kaolin, into geopolymer materials with desirable properties by chemically reacting with an alkaline activator solution [11,12,13]. Past research had concluded that the geopolymerization process typically involves dissolution, polycondensation, and curing stages. The dissolution stage occurs when kaolin is mixed with an alkaline activator forming a slurry. The alkaline activator initiates the dissolution of the precursor, leading to the release of silica and alumina ions. Then, the dissolved silica and alumina ions undergo polycondensation reactions [14], which involve the formation of covalent bonds between the tetrahedral and octahedral units. The resulting product is a three-dimensional network of linked tetrahedral and octahedral units, forming the geopolymer materials. In the curing stage, the geopolymer is set to cure or harden. During this time, the geopolymer material undergoes further chemical and physical changes, such as the formation of additional covalent bonds and the development of its final mechanical properties. The specific processing conditions, including the type of aluminosilicate source materials [15], the concentration of the alkaline activator [16], and the curing conditions [17], can significantly influence the properties of the geopolymer. By carefully controlling these factors, it is possible to tailor the properties of geopolymer materials to meet specific application requirements, such as mechanical strength, thermal stability, and chemical resistance.

From the idea of geopolymer production, geopolymer-based ceramics are introduced as an alternative in the field of ceramic production due to their ability to offer enhanced thermal stability, chemical resistance, and mechanical strength [18], and most importantly, they required low sintering temperatures compared to conventional ceramic fabrication, where the sintering temperature goes up to 1600 °C. Several other problems are also found in using the conventional method, such as prolonged heating time, irregular grain growth, and furnace contamination. Therefore, the geopolymerization method has been adapted to the fabrication of geopolymer-based ceramics to progress the application of the geopolymer technology. Apart from geopolymerization, the sintering temperature also affects the properties of a geopolymer when heat is applied to geopolymer bodies. Therefore, in this study, the effects of the sintering mechanism are investigated to study how it affects the properties and crystallization of the geopolymer ceramics produced.

## 2. Experimental Method

### 2.1. Materials

Kaolin is a clay mineral mainly containing a chemical composition of Al_2_Si_2_O_5_(OH)_4_. In this study, kaolin was used as a starting material for geopolymerization. The kaolin used was supplied by Associated Kaolin Industries Malaysia as Si-Al source materials, where the large compounds found are SiO_2_ and Al_2_O_3_. Table 1 shows the chemical composition of kaolin obtained by X-ray fluorescence (XRF).

A mixture of 12 M sodium hydroxide (NaOH) and sodium silicate (Na_2_SiO_3_) as an alkaline activator was used to activate the source material. The ratio of Na_2_SiO_3_/NaOH was set at 0.24. The sodium hydroxide caustic soda micro-pearls with a purity of 99% were supplied by Formosoda-P from Taiwan. Meanwhile, sodium silicate (Na_2_SiO_3_) was provided by South Pacific Chemicals Industries Sdn. Bhd. (SPCI), Pahang, Malaysia with the chemical composition of H_2_O (60.5%), SiO_2_ (31.1%), and Na_2_O (9.4%).

### 2.2. Sample Preparation

Figure 1 shows the overall process of synthesizing geopolymer ceramics. The influence of sintering temperature on the green bodies in the production of kaolin-based geopolymer ceramics was investigated by systematically varying the sintering temperature within the range of 200 °C to 1200 °C. The aluminosilicate source, kaolin was mixed with an alkali activator at a solid-to-liquid ratio of 1.0 to activate the source material. The ratio of the alkali activator and molarity of NaOH was fixed at 0.24 and 12 M, respectively. The solution needed to be prepared 24 h before it was used to obtain a homogeneous solution. The mixture was then cured at 80 °C in an oven for 24 h. To obtain the fine powder, the kaolin geopolymer was crushed using a mechanical crusher and sieved by using a 150 µm sieve. Subsequently, the compacted geopolymer (86 MPa, 2 min) underwent a sintering process at various temperatures (200 °C, 400 °C, 600 °C, 800 °C, 1000 °C, and 1200 °C), with a soaking duration of 180 min and a heating rate of 5 °C/min. Previously, Ahmad et al. [19] had studied the sintering profile of geopolymer-based ceramics in the temperature range of 900 °C to 1200 °C, and obtained 1200 °C as the maximum sintering temperature. Therefore, in this research, a thorough procedure is prepared from low sintering temperature of 200 °C to study the evolution of the properties of geopolymer-based ceramics. Figure 2 shows the sintering profile for the fabrication of geopolymer ceramics.

To examine the influence of temperature on the properties of geopolymer ceramics, an extensive characterization of all prepared samples was conducted. The mechanical properties, specifically the flexural strength of the geopolymer ceramics, were evaluated. The three-point bending method, adhering to the guidelines of ASTM C-1163b, was implemented with a support span length of 30 mm and a crosshead speed of 0.3 mm/min. The morphology of the geopolymer bodies was studied by using a JEOL JSM-6460LA scanning electron microscopy (SEM). The microstructural development of the geopolymer and geopolymer ceramics as well as identified reacted/unreacted regions, porosity, and cracks were examined to study the effect of the sintering mechanism. To create a conductive layer, the sample was coated with gold using a JEOL JFC 1600 model auto fine coater. This data is supported by the results of the Synchrotron radiation X-ray tomographic (SXTM), used to study the porosity of the geopolymer ceramics. The test was carried out at the Synchrotron Light Research Institute (SLRI), Thailand.

The phase composition of the samples was determined by an XRD 6000, SHIMADZU diffractometer. The sample was ground into a fine powder by using a ring mill small enough to ensure that the X-ray can penetrate the sample and generate a diffraction pattern. The sample was then mounted onto a sample holder and evenly distributed. To ensure the diffraction pattern is accurate and reproducible, the sample was aligned so that the beam was perpendicular to the surface of the sample. The data were collected over a range of 10° to 65°.

## 3. Results and Discussions

### 3.1. Mechanical Properties of Geopolymer-Based Ceramics

The effect of thermal treatment on the mechanical properties of geopolymer-based ceramics is shown in Figure 3, Figure 4, Figure 5 and Figure 6, based on the mean of the result of the flexural strength, density, shrinkage, and water absorption of the sintered geopolymer ceramics, respectively. The error bars in the graph represent the standard deviations of the data. Higher sintering temperatures typically lead to denser ceramic materials due to increased grain diffusion and densification. As a result, higher sintering temperatures generally lead to higher flexural strength as well, as the denser material is better able to resist bending and deformation under mechanical stress. Upon increasing the temperature, the flexural strength increases from 7.18 MPa for unsintered geopolymers, caused by crystal growth, which strengthens the materials and reduces their susceptibility to deformation and cracking. When geopolymer-based ceramics are exposed to 1000 °C, a high flexural strength of 53.5 MPa is recorded. When further sintered at 1200 °C, the strength achieves its maximum flexural strength of 78.9 MPa, which is caused by the densified and crystallized matrix and the enhanced fiber/matrix interface bonding as the sintering mechanism occurs [20].

Figure 4 presents the density of the unsintered and sintered geopolymer at various sintering temperatures of 200 °C, 400 °C, 600 °C, 800 °C, 1000 °C, and 1200 °C. As displayed, along with the increments of the sintering temperature, the density of the geopolymer-based ceramics also changes in increments. At 1200 °C, the density recorded the highest value of 2.56 g/cm^3^, compared to a geopolymer sintered at 200 °C (1.61 g/cm^3^) and an unsintered geopolymer (1.45 g/cm^3^). The increase in the density of the geopolymer-based ceramics is related to the greater grain boundary diffusion which promotes densification of the ceramics, thus increasing the strength [21]. High sintering temperature crystallize the inner structure of the geopolymer-based ceramics, thus enhancing the mechanical performance of the ceramics. The density of the ceramics is dependent on the sintering temperature. The sintering temperature plays a significant role in the densification process. Exposure to a high temperature eliminates the pores between particles and also facilitates fast grain growth during the sintering mechanism [22].

When sintered at 200 °C, the occurrence of dehydration stages leads to shrinkage and deformation. Linear shrinkage was defined as the reduction percentage in the thickness of the central part of the powder compacts. This can be attributed to capillary contraction induced by the shape of water within micro and nanopore solutions. Generally, linear shrinkage increases as the sintering temperature increases. Increasing the sintering temperature led to greater shrinkage in the materials, as the higher temperatures cause more particle rearrangement and compaction. The body of the ceramics achieved its maximum shrinkage of 29.91% when exposed to 1200 °C. A study by Jaya et al. [23] stated that sintering above 1200 °C is not suitable as the geopolymer palette starts to melt as the temperature rises to 1250 °C, turning the color from milky white to brown. This happens due to the exceeding of the melting point of the geopolymer ceramics.

As the geopolymer undergoes sintering at elevated temperatures, the structure of the material is modified. These modifications encompass the elimination of water molecules and the formation of a new crystalline phase embedded within the geopolymer matrix. The surface energy of incoherent particles decreases during the sintering mechanism, leading to a decrease in the overall surface area. The application of high temperatures during the sintering process induces structural rearrangement and the development of crystalline phases, including nepheline (NaAlSiO_4_), kalsilite (KAlSiO_4_), and mullite (3Al_2_O_3_·2SiO_2_) [24]. This sintering mechanism leads to enhanced material density and facilitates the growth of grains through diffusion. As the liquid flows between particles, shrinkage takes place, exerting greater pressure at the contact areas and prompting the material to relocate, resulting in closer proximity between particle centers.

As for the water absorption (WA), the decreasing trend is possibly due to the densification process of the geopolymer matrix in the ceramic materials as the sintering temperature increases. When sintered above 1200 °C, the geopolymer-based ceramics absorbed 0.23% of water in 24 h compared to unsintered and a geopolymer sintered at 200 °C which absorbed 28.63% and 28.55% of water, respectively. The fluctuation in water absorption may be due to the elevated temperature, resulting in the closing of some open pores [25], thus reducing the amount of available pore space for water to be absorbed. This phenomenon causes the rate of water absorption to decrease while increasing the pore strength. The low water absorption exhibits a better property for geopolymer-based ceramics as it contributes to an increase in strength and creates barriers to the formation of cracks and voids [26].

### 3.2. Morphology Analysis and Porosity of Geopolymer-Based Ceramics

The SEM micrograph provides a visual representation of the unsintered geopolymer, illustrating the scanning image obtained during the activation of kaolin with the alkali activator. The plate-like morphology of unreacted kaolin in geopolymer ceramics can be observed in Figure 7. It has a unique morphological structure characterized by its elongated, plate-like particles, which function to reinforce the geopolymer matrix, thus increasing the strength and toughness of the resulting geopolymer-based ceramics.

Figure 8 shows the SEM micrograph of unsintered and sintered kaolin-based geopolymer ceramics at various sintering temperatures of 200, 400, 600, 800, 1000, and 1200 °C. All of the samples have been imaged at a fracture section of the geopolymer-based ceramic. Only a few micro-level cracks were visible on the sample surface when exposed to a higher temperature. Following the heat treatment, as an effect of the grain coarsening phenomenon, the small pores within the matrix gradually disappear, and the larger pores form. There is no significant change in the microstructure of geopolymer when exposed to temperatures of 200 °C, 400 °C, and 600 °C. During heating from room temperature (RT) to 200 °C, the water content in the geopolymer evaporates causing weight loss and minimal shrinkage. The shrinkage and deformation could be attributed to the capillary contraction induced by the escape of water from the pores [27]. However, as the geopolymer sintered at 800 °C, it can be observed that the geopolymer matrix starts to become more dense. This phenomenon can be attributed to the dehydroxylation stages, during which condensation and polymerization between T-OH groups leads to the escape of water and subsequent shrinkage at high sintering temperatures. Sintering above 800 °C causes the flexural strength to increase [28]. The thermal analysis revealed the occurrence of the sintering process, which was evident from the observed shrinkage resulting from crystal coarsening. In accordance with Li et al. [29], with an increase in sintering temperature, there is a promotion in the growth of sintering necks and sintering densification. This, in turn, leads to an enhancement in the flexural strength of the geopolymer-based ceramics. Subsequent sintering to 1200 °C induces the formation of a molten amorphous glass phase, promoting further sintering and densification. This process significantly contributes to the maximum shrinkage observed. Small pores form from an amorphous geopolymer into nepheline geopolymer-based ceramics.

The sintering process above 1000 °C resulted in the development of a glassy morphology, signifying the closure of the majority of the small pores and the attainment of ceramic densification [30]. Figure 9 shows the proposed scheme on the grain growth of the geopolymer-based ceramics during the sintering mechanism which leads to the densification of the geopolymer matrix. During the initial stage of sintering, the coalescence and orientation of particles reduces the coordination number around pores and alters the balance of surface tension around pore surfaces, which causes the closure of pores. As the sintering temperature increases, the grain grows and leads to the formation of densified geopolymer-based ceramics.

At a high temperature, the ceramic structure undergoes significant structural changes, where the porosity and mechanical properties are affected. As the geopolymer is exposed to high sintering temperatures, the geopolymer structure begins to collapse and form cracks. This may create new pores and increase the overall porosity of the material. While increasing porosity is expected to decrease mechanical strength, there are cases where the mechanical strength could be increased as the porosity increases. This is due to the function of the pores that can act as stress concentrators, which helps in dissipating stress and preventing the initiation and propagation of cracks [31].

The porosity of geopolymer-based ceramics can be observed in Figure 10 using data from the XTM technique, where it is shown that the total number of pores is highest at 6.61% when sintered at 1200 °C, at which point they mainly consist of closed pores. This is due to the sintering mechanism where, as the sintering temperature increases, the small pores merge forming a larger pore due to moisture hydration. Even though the total number of pores is highest at this temperature, the closed pores present contribute to the high density, thus improving the strength of the geopolymer-based ceramics. Contrary to sintered ceramics, the unsintered geopolymer happens to have the lowest number of pores with 0.2%. This result is thus related to the SEM micrograph obtained.

### 3.3. Phase Analysis of Geopolymer-Based Ceramics

Figure 11 shows the XRD pattern of unsintered kaolin-based geopolymer. The presence of zeolite (Z), kaolinite (K), and quartz (Q) were detected in the kaolin-based geopolymer. The kaolinite (Al_2_SiO_5_(OH)_4_) and quartz (SiO_2_) represent inherent mineral constituents of kaolin, whereas zeolite typically crystallizes through the activation of kaolin with the alkali activator, originating from the transformation of amorphous aluminosilicate gel. Geopolymerization is initiated by the combination of NaOH and Na_2_SiO_3_ solutions, leading to the dissolution of aluminosilicate minerals in an alkali activator. The dissolved components undergo a series of processes involving nucleation, growth, and polymerization before ultimately solidifying through polycondensation. The alteration in crystallographic composition was evidenced by the disappearance of the amorphous phase hump upon heat treatment, indicative of the transition from an amorphous to a crystalline state.

Figure 12 provides a characterization of geopolymer-based nepheline ceramics, encompassing sintered samples exposed to different temperatures of 200 °C, 400 °C, 600 °C, 800 °C, 1000, and 1200 °C. At the temperatures of 200 °C, 400 °C, and 600 °C, amorphous humps and reflections from quartz present between the range of a 20° to 40° diffraction angle (2θ) were introduced by the raw kaolin. Sharp peaks in the reflection of nepheline (NaAlSiO_4_) start to appear when the geopolymer-based ceramics are sintered at a temperature of 800 °C. The absence of muscovite (KAl_2_(AlSi_3_O_10_)(OH)) could be attributed to the exposure of the kaolin-based geopolymer to high temperatures during the sintering process. As the sintering temperature increased to 800 °C, the intensity of the peaks increased, and the amorphous phase diminished, indicating the initiation of geopolymer crystallization [32].

Sintering at 1000 °C and 1200 °C resulted in the formation of mullite (3Al_2_O_3_·2SiO_2_), which exhibits excellent thermochemical stability. The XRD pattern recorded for 800 °C, 1000 °C, and 1200 °C showed an incremental rise in the intensity of the nepheline peaks as the sintering temperature increased. Nepheline contributes to the strength and stability of ceramics. It also helps in improving their resistance to heat and erosion. Furthermore, nepheline can impact the physical and mechanical properties of ceramics, such as their density, hardness, and thermal conductivity [33]. Overall, incorporating nepheline into a kaolin-based geopolymer can improve its performance and make it more suitable for a range of applications.

## 4. Conclusions

Geopolymer-based ceramic materials have numerous potential uses, such as building materials, refractories, high-temperature coatings, and even as an alternative to conventional cement-based materials. The spectrum of potential uses for geopolymer materials is projected to broaden with continued study and development, making them a more crucial component of contemporary engineering and building.

From the obtained results, it can be inferred that the sintering process has an impact on the characteristics and properties of geopolymer-based ceramics. The phase analysis conducted by using XRD indicates that the results align with the compressive strength and the morphology analysis. By detecting the major crystalline components present in the geopolymer-based ceramic systems, phase analysis aids in explaining the properties of the geopolymer. As the geopolymer is sintered at high temperatures above 800 °C, the XRD pattern shows an incremental increase in the intensity. Additionally, from the SEM micrograph of the fracture section of the geopolymer-based ceramics, the morphology analysis of geopolymer-based ceramics sintered at 1200 °C reveals a denser surface appearance compared to the unsintered geopolymer. Furthermore, sintering the ceramics at a temperature of 1200 °C enhances their mechanical performance as they achieved a maximum flexural strength of 78.9 MPa, a peak density of 2.56 g/cm^3^, and their lowest water absorption at 0.23%.

## Figures and Tables

**Figure 1 materials-16-05853-f001:**
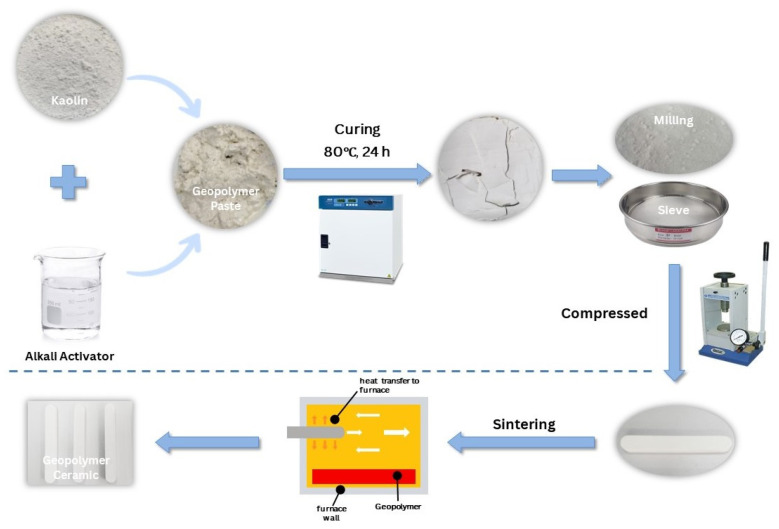
The overall process of synthesizing geopolymer ceramics.

**Figure 2 materials-16-05853-f002:**
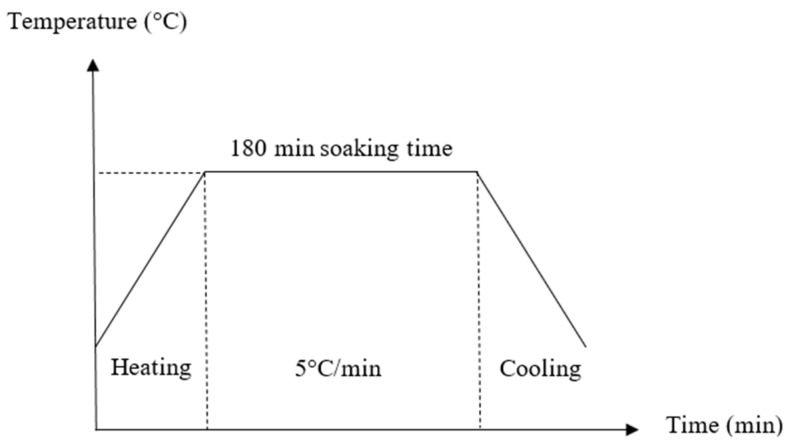
Sintering profile for fabrication of geopolymer ceramics.

**Figure 3 materials-16-05853-f003:**
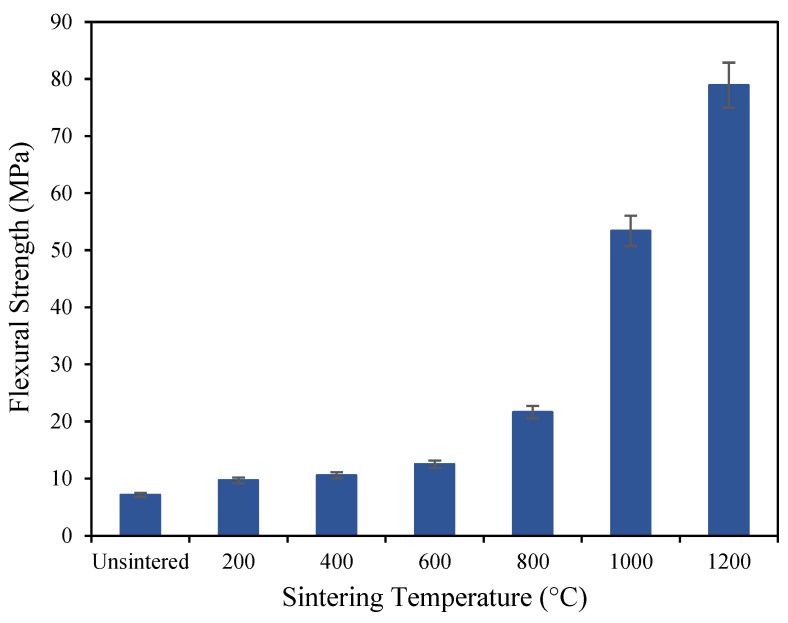
Flexural strength of unsintered and sintered geopolymer ceramics.

**Figure 4 materials-16-05853-f004:**
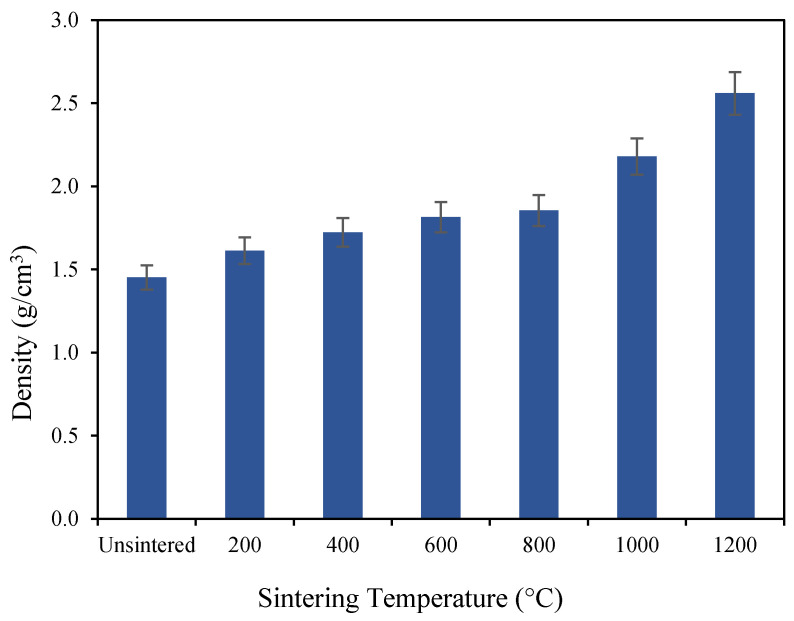
Density of unsintered and sintered geopolymer ceramics.

**Figure 5 materials-16-05853-f005:**
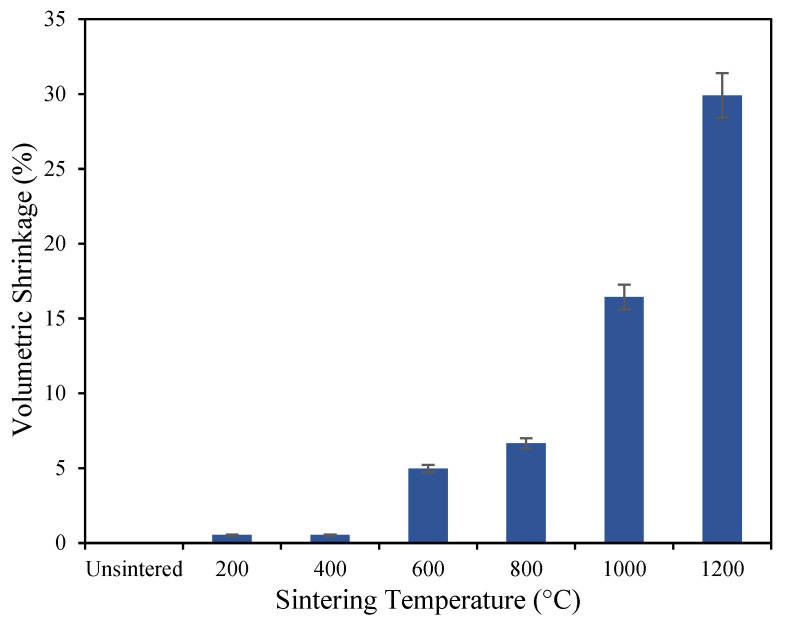
The shrinkage of geopolymer-based ceramics after the sintering process.

**Figure 6 materials-16-05853-f006:**
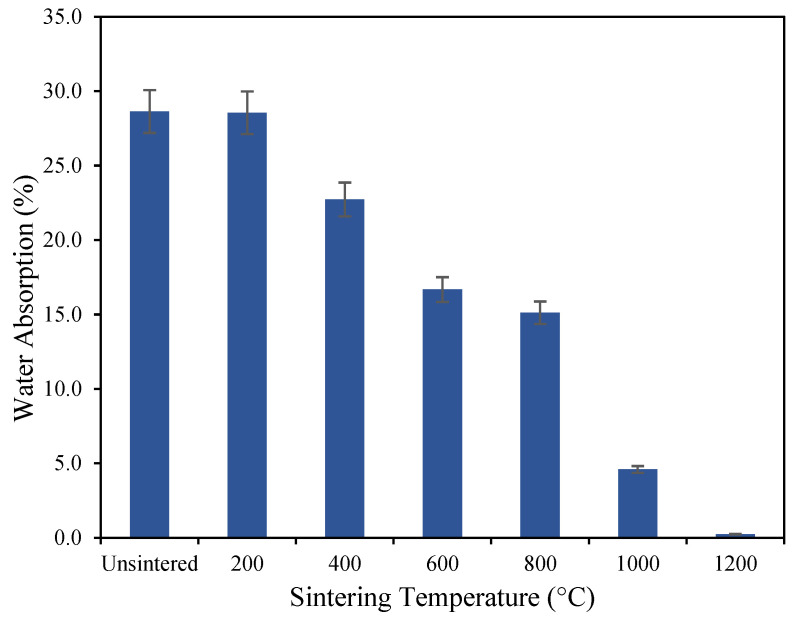
The water absorption of geopolymer-based ceramics.

**Figure 7 materials-16-05853-f007:**
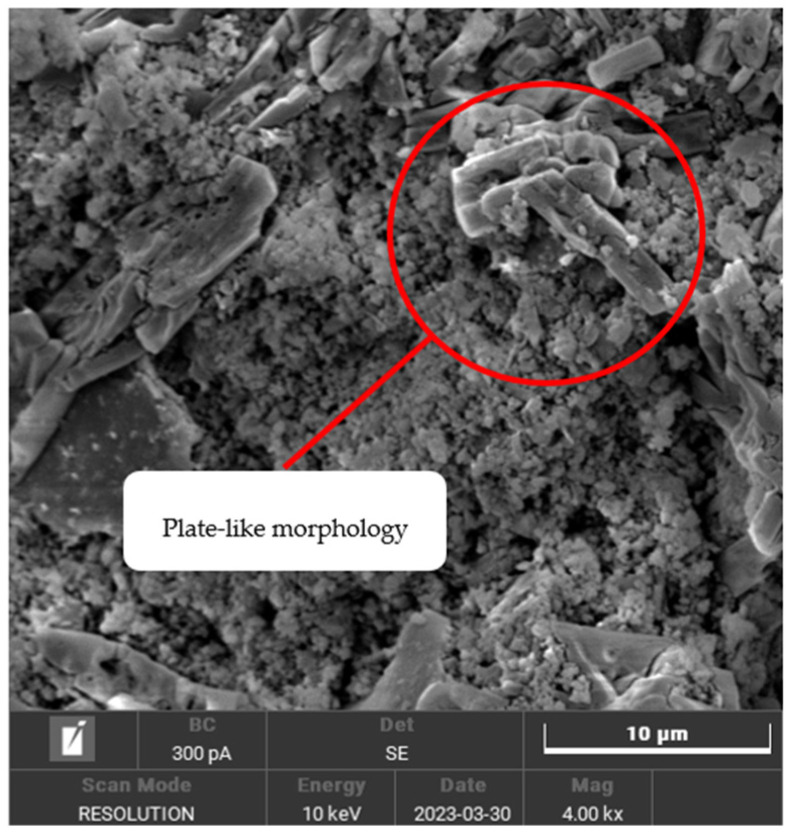
SEM micrograph of unreacted plate-like kaolinite at 4000× magnification.

**Figure 8 materials-16-05853-f008:**
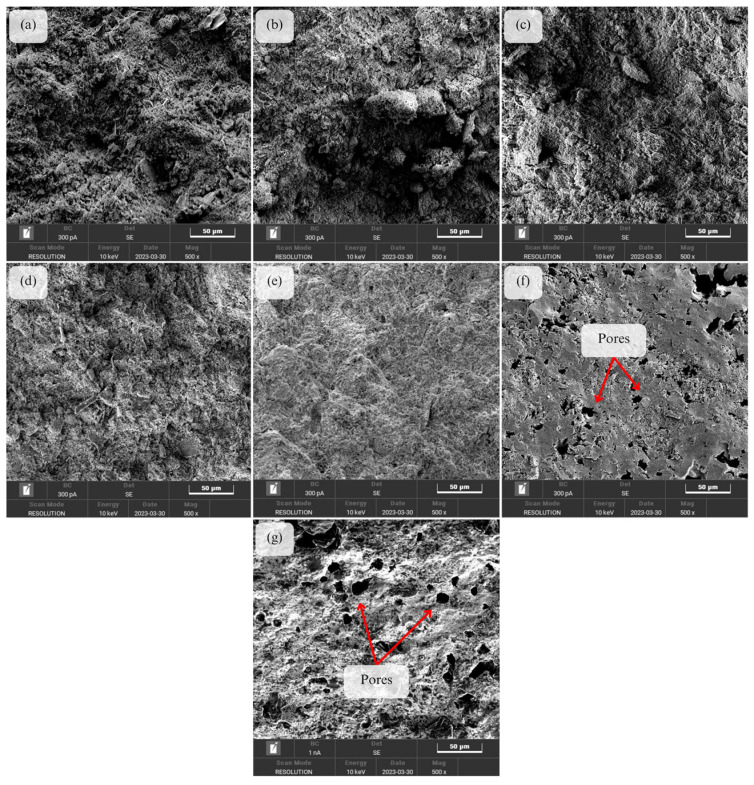
SEM micrograph of ceramic samples for (**a**) unsintered and sintered samples at (**b**) 200 °C, (**c**) 400 °C, (**d**) 600 °C, (**e**) 800 °C, (**f**) 1000 °C, and (**g**) 1200 °C.

**Figure 9 materials-16-05853-f009:**
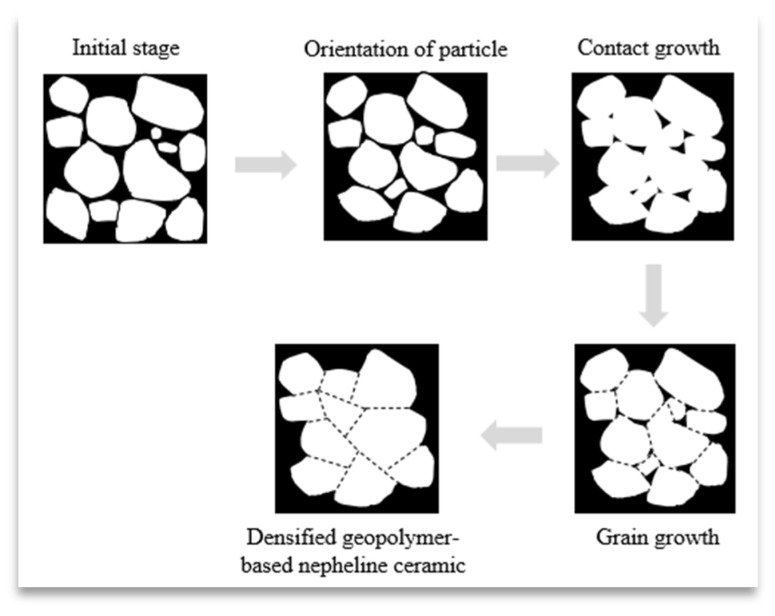
Proposed scheme of the grain growth during the sintering mechanism of geopolymer-based ceramics.

**Figure 10 materials-16-05853-f010:**
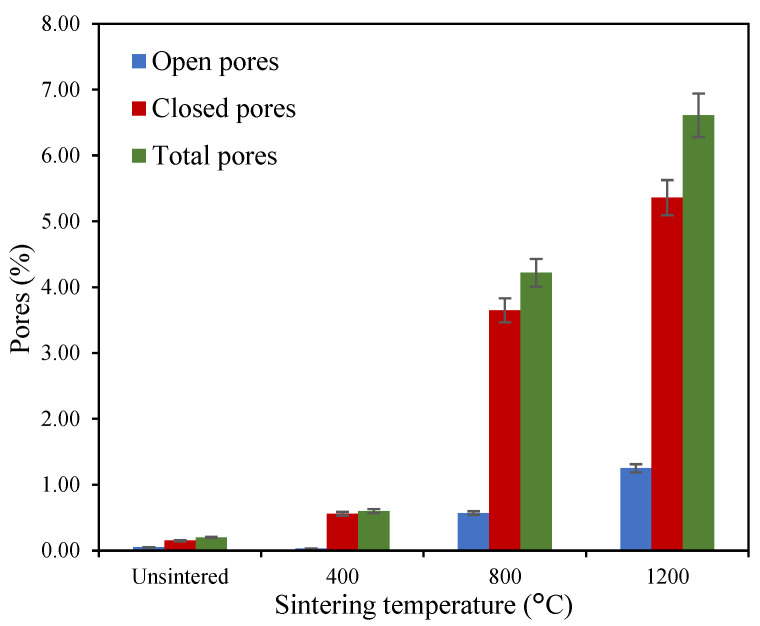
The porosity of unsintered and sintered geopolymer-based ceramics from XTM.

**Figure 11 materials-16-05853-f011:**
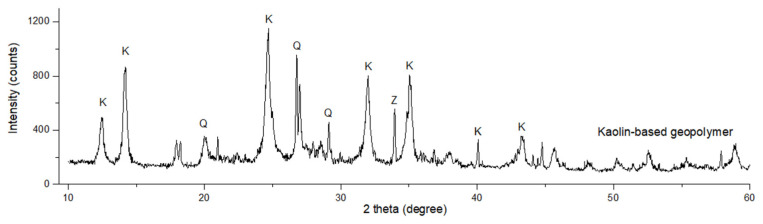
XRD patterns of unsintered kaolin-based geopolymer (K = kaolinite, Q = quartz, Z = zeolite).

**Figure 12 materials-16-05853-f012:**
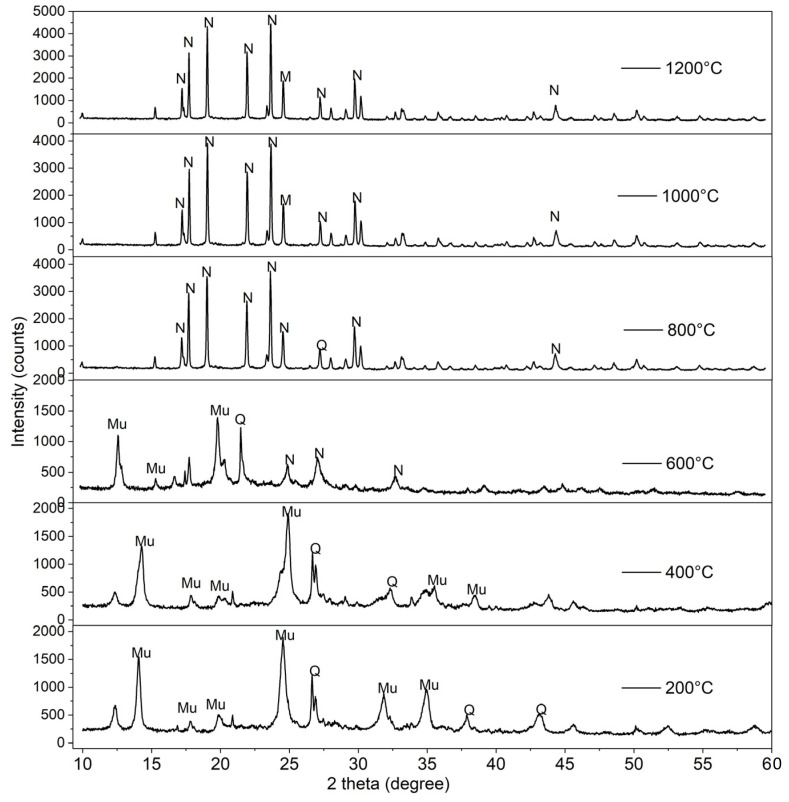
XRD patterns of sintered kaolin-based geopolymer at 200 °C, 400 °C, 600 °C, 800 °C, 1000 °C, and 1200 °C (Mu = muscovite ICDD# 05-0652, Q = quartz ICDD# 46-1045, N = nepheline ICDD# 35-0424).

**Table 1 materials-16-05853-t001:** Chemical composition of kaolin (wt.%) obtained by X-ray fluorescence.

Element	SiO_2_	Al_2_O_3_	K_2_O	Fe_2_O_3_	TiO_2_	MnO_2_	ZrO_2_	LOI
Wt. (%)	54.0	31.7	6.05	4.89	1.14	0.11	0.10	1.74

## Data Availability

Not applicable.

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
