# Peer review of "Effect of the Sintering Mechanism on the Crystallization Kinetics of Geopolymer-Based Ceramics"

_materials, 2023, doi:10.3390/ma16175853_

Round 1
Reviewer 1 Report
Authors N. B. Mustapa at al. present in their manuscript the results of investigation where they prepared geopolymer from kaolin clay and additionally heated/fired at temperatures up to 1200oC. The samples were characterized by many analytical methods.
1. In the Introduction section the authors mention that for conventional sintering process high temperatures are used up to 1600oC. It is true for some ceramics (Al2O3, ZrO2, ….) but not for aluminosilicate ceramics. Namely, the authors describe in the manuscript the results of investigation of aluminosilicate materials.
2. Line 123: The authors write: “…where the large element found is SiO2 and Al2O3.“ Actually, SiO2 and Al2O3 are not elements. This should be corrected.
3. Lines 128-129: The authors write: “The ratio of the alkali activator and molarity of NaOH is fixed at 0.24 by mass and 12 M, respectively“. I do not understand this sentence. What was the concentration of NaOH in aqueous solution and concentration of Na2SiO3?
The same sentence is repeated in the next subsection in lines 138-139. Why?
4. In the Fig. 1 there is a pressure presented in tons. Usually, a pressure is presented in Pa (MPa). Correct it!
5. Line 174. The authors write: “The effect of thermal on the mechanical properties…“ This is an incomplete sentence.
6. Line 177: The authors write: “….increased particle diffusion and ….“ Are the authors sure that during sintering process “particle diffusion” occurs?
7. The explanation of adsorbing water (lines 210-217) is not clear. This paragraph should be rewritten.
8. SEM micrograph is already an image (line 221).
9. Line 28: The authors write “ SEM micrograph of unreated plate-like kaolin…“ There is a typo in ”unreated”. These are kaolinite particles. The authors should differentiate between kaolin and kaolinite. And in the Fig. 6 there is not “Plate-like structure” but plate-like morphology.
10. Line 231: The authors write: “The specimen has been imaged from a fracture … “ Did the authors prepare/analyze only one specimen?
11. Lines 233-234: The authors write: “…a notable grain coarsening phenomenon was observed“ Honestly speaking, this phenomena can not be observed in Fig. 7.
12. Lines 237-238: The authors write: “Figure 8 shows the grain growth of the geopolymer-based…“ This is not true. Fig 8 only shows cross-section of the fracture of the sample where large pores can be seen and the scheme how the authors proposed grain growth. Correct the claim. In addition, in this Fig the authors also mention “grain boundary development”. However, there are no grain boundary that can be seen.
13. Line 249: The authors write: “During heating, the water content in the geopolymer evaporates“ The authors should specify in which temperature range water evaporates.
14. Line 285: The authors write: “…and quartz amorphous phases“ What does this mean? XRD can not distinguish between one are more amorphous phases. Reflections in diffractograms in Fig. 10 should be identified and presented in the figure. Which XRD cards were used for identification of the present phases?
15. The authors write: “suggest that there is a significant amount of amorphous phase present in the kaolin sample“ In general, amorphous phase has much higher solubility in alkali activation reaction in comparison to crystalline phase(s). Thus, it is important to know how much of amorphous phase was in the starting mixture. Is it possible, that the authors asses this amount?
Author Response
Dear editor,
We express our gratitude to both you and the reviewers for dedicating your valuable time to review our paper and share your valuable insights. Your comments have played a crucial role in enhancing the current version of the manuscript. The authors have diligently examined each comment and made every effort to address them comprehensively. We aim to meet your high standards through careful revisions of the manuscript. If you have any further constructive comments, we would be delighted to receive them.
Please find our detailed responses to each point below, with all modifications highlighted in yellow.
Thank you.
Regards,
Nur Bahijah Mustapa, MSc. Students,
Faculty of Mechanical Engineering Technology,
University Malaysia Perlis (UniMAP).

Reviewer 2 Report
Dear authors, please consider some observations below reported:
2.1 Materials
Change …….where the large element found is SiO2 and Al2O3…….. in …….where the large elements found are SiO2 and Al2O3……..
2.2 Samples Preparation
Whith this sentences, in particular the last:
…..Subsequently, the compacted geopolymer will undergo a sintering 143 process at various temperatures (200 °C, 400 °C, 600 °C, 800 °C, 1000 °C, and 1200 °C), with a soaking duration of 180 minutes and a heating rate of 5 °C/min. This meticulous procedure aims to yield the desired outcome of geopolymer ceramics as the final product…..
what the authors mean? The authors sholud better explain the choice of the thermal cycle (probably others test have been performed previusly) and the meaning of the desired outcome of geopolymer ceramics as the final product
3. Findings and Discussions
In my opinion is better to change the title in Results and Discussion
3.1 Mechanical properties of geopolymer-based ceramics
How many samples have been measured for each parameter? I suppose that each value reported in Figures 3, 4 and 5 are the mean of different measurements?! Please, indicate this.
In each bar of the histograms in Figures 3, 4 and 5, what does the vertical bar mean? The error, the standard deviation? It must be specified.
In general, in this paragraph, the authors should compare the results obtained with analogous parameters of ceramics obtained by simple sintering of powder (in other words, by the traditional ceramic process), in order to make an objective comparison. Possibly say which ceramic product it is similar to, i.e. porcelain stoneware, single-fired, double-fired, etc.
3.2 Morphology analysis and porosity of geopolymer-based ceramics
In the sentence….. Figure 7 shows the SEM micrograph of unsintered kaolin-based geopolymer ceramic when sintered at various sintering temperatures of (200, 400, 600, 800, 1000, and 1200 °C)…………..I suggest to delate “unsintered”
In figure caption…..Figure 7. SEM micrograph of a) Unsintered b) 200 °C, c) 400 °C, d) 600 °C, e) 800 °C, f) 1000 °C and 250 g) 1200 °C….I propose to write: Figure 7. SEM micrograph of a) unsintered sample and sintered samples at b) 200 °C, c) 400 °C, d) 600 °C, e) 800 °C, f) 1000 °C and g) 1200 °C ….…
In the sentence … The porosity of geopolymer-based ceramics can be observed in Figure 9 using data from XTM, where it is shown that the total pores are the highest when sintered at 1200 °C which is at 6.61% where it mainly consists of closed pores…… I suggest to write XTM technique
3.3 Phase analysis of geopolymer-based ceramics
In Figure 10 the authors should provide for the association of the peaks to the crystalline phases, indicating the number of identification cards
4. Conclusions
Implement this paragraph with the comparison of the properties of the samples studied here with traditional ceramics. That is, to take up a brief conclusion from the additions suggested in the part of Results and Discussion.
Author Response

(The authors gave the same response as above.)

Reviewer 3 Report
Nice paper dealing with the synthesis and characterization of geopolymers (ceramics) by various techniques. It is suitable for publication after minor revision.
Comments
1. Did the authors investigate or have an intention to investigate (di)electrical properties of samples? It could widen the application of the material and help to understand reaction mechanisms. Some of the papers dealing with the electrical properties of geopolymers are published in Materials
Górski, Marcin, et al. "Electrical Properties of the Carbon Nanotube-Reinforced Geopolymer Studied by Impedance Spectroscopy." Materials 15.10 (2022): 3543.
Istuque, Danilo Bordan, et al. "Impedance Spectroscopy as a Methodology to Evaluate the Reactivity of Metakaolin Based Geopolymers." Materials 15.23 (2022): 8387.
2. The authors show that sintering at 1200 oC significantly enhances the material properties. But may be 1300 or 1400 oC would be even better? Did the authors try the elevated temperatures? Or maybe they give reasonable reply why it is not feasible?
3. One of the conventional approaches to study new materials is the IR spectroscopy. IR spectroscopy can also help and give information about the presence (quantity) of water, CO- OH-groups, etc. with the sintering procedure. In the presented paper changes with temperature are described mainly suggestively, IR measurements could support the theoretical proposals.
Author Response

(The authors gave the same response as above.)

Reviewer 4 Report
In this manuscript, authors studied the microstructure and mechanical properties of the geopolymer obtained at different sintering temperatures. The geopolymer consisted of kaolin and sodium silicate as a precursor and an alkaline activator.
Generally, the work is good and the results are interesting for the industrial manufacturing.
Article was prepared well and corresponds to the subject of the Journal.
I have some notices.
- In Findings and Discussions. Page 5, lines 193-194. The authors found that the ceramic reached a maximum shrinkage of 29.91% at 1200°C. The article does not explain how expedient a further increase in temperature is?
- Page 5, lines 199-201. The authors explain that the application of high temperatures during the sintering process causes a structural rearrangement, resulting in the formation of crystalline phases, including nepheline calcite and mullite. This paragraph needs to include a reference to the literature or its XRF data.
- Page 8, lines 256-257. The authors explain: «While in some cases increasing porosity is expected to decrease the mechanical strength, there are cases where the mechanical strength could be increased as the porosity increase. This is due to the function of the pores that can act as stress concentrators, which helps in dissipating stress and preventing cracks propagation». This needs to include a reference to the literature.
- It is not clear why the authors did not study the density and porosity of the samples using the Archimedes technique? There are some inconsistencies in the results obtained, i.e. high water absorption and low porosity (from XTM) in samples made at low sintering temperatures.
Author Response

(The authors gave the same response as above.)

Reviewer 5 Report
The paper presents interesting research, describing the new technological process of ceramic materials and in my opinion, it can be published after minor revision. Detailed comments are below.
Generally, the authors should compare the results of their research with the results presented in the available literature for ceramics sintered in the classical technology, thus showing the advantages of their method of obtaining.
- lines 26-33: “The uprising demand for ceramics …. process is not much detailed.” these sentences should be placed in the Introduction section.
- line 97: “…materials. In…” uneccessary double space.
- Line2 124, 125: “X-Ray” should be “X-ray”.
- lines 147, 151: “ceramic” should be “ceramics”.
- lines 157, 158, 161, 162: “Scanning Electron Microscopy”, “Auto Fine Coater”, “Synchrotron” use lowercase letters.
- line 190: “…solutions. Generally,…” uneccessary double space.
- Fig.3, Fig.4: incorrect degree symbol – there is “0” in superscript.
- lines 212, 213: “0.236%” , “28.637%, “28.555%” I don't think such precision is necessary
- Fig.6: put the scale on the SEM image.
- line 229: “ceramic” should be “ceramics”.
- Fig.7: put the scale on the SEM images.
- line 250: “SEM micrograph of…”, should be “SEM micrograph of ceramic samples...”.
- line 268: “ceramic” should be “ceramics”.
- Fig.8 put the scale on the SEM image. Moreover, from the SEM image I can see that it is an image of the microstructure of the sample surface, while in the case of the previous images there is a cross section of the sample. It would be good to put that in the description.
- lines 283, 285, 303: – The authors write: "XRD spectra", “XRD diffraction patterns”. A XRD spectrum shows an event (e.g., intensity) depending on energy, a diffractogram shows an event depending on scattering angle.
- line 290: “ceramic” should be “ceramics”.
- line 286: “…Figure 10 The…” missing dot.
- Fig.10 put “(-)” after “Intensity”. Moreover, there is no matching of the experimental results to the pattern. It should be supplemented.
Author Response

(The authors gave the same response as above.)

Round 2
Reviewer 1 Report
The authors have improved the manuscript, however there are still lacks that should be improved/corrected.
1. In my previous review I suggested that the authors express pressure in Pa. Their response was: “However, the laboratory press used to compact the geopolymer are in the unit of metric tons.“ I insist that the authors calculate pressure according to applied force and surface of the pellets and express it in Pa.
2. The authors are suggested to divide fig. 9 into 9a and 9b and write in figure caption that fig. 9b shows proposed scheme of grain growth. I must stress that the authors do not present any micrograph of their sintered samples where grains can be seen.
3. I suggested in previous review that the authors mention in the manuscript which XRD cards were used for identification of the present phases, however there is no this information in the reviewed manuscript. Why? In addition, the authors write a formula only for nepheline, please write formula also for other phases that were identified, for example muscovite, berlinite, mullite, zeolite etc. I also mentioned that the authors estimate a share of amorphous phase in kaolin, but they did not?
4. Line 313-314; The authors write “The XRD spectra reveal distinct peaks between 20° and 40°, indicating the presence of an amorphous phase in the kaolin“. The authors should label these peaks in XRD pattern of kaolin that supposedly can be ascribe to amorphous phase!
5. Lines 322-323: The authors write “Analysis of the XRD patterns indicated a decrease in peak size with increasing sintering temperature, …“ Honestly speaking, I observe opposite trend in intensity of reflection in XRD patterns in the Fig. 11. For example, the y axis for the starting material extends up to 1950 counts while for the sample sintered at 1200oC extends to 2250 counts. In addition, more intensive peaks in XRD pattern suggested that the (poly)crystalline analyzed phase has more ordered crystal structure and larger grains.
6. Line 324: “…the intensity of the amorphous phase is reduced “. This is not correct sentence.
7. Lines 332-333: “Upon sintering the geopolymer-based ceramic at temperatures above 800 °C, a pronounced increase in intensity is detected “ I do not understand this sentence.
Author Response
Dear editor,
We express our gratitude to both you and the reviewers for dedicating your valuable time to review our paper and share your valuable insights. Your comments have played a crucial role in enhancing the current version of the manuscript. The authors have diligently examined each comment and made every effort to address them comprehensively. We aim to meet your high standards through careful revisions of the manuscript. If you have any further constructive comments, we would be delighted to receive them.
Please find our detailed responses to each point below, with all modifications highlighted in green.
Thank you.
Regards,
Nur Bahijah Mustapa, MSc. Students,
Faculty of Mechanical Engineering & Technology,
University Malaysia Perlis (UniMAP).

Reviewer 4 Report
No comments.
Author Response
Dear editor,
We express our gratitude to both you and the reviewers for dedicating your valuable time to review our paper and share your valuable insights. Your comments have played a crucial role in enhancing the current version of the manuscript. We aim to meet your high standards through careful revisions of the manuscript. If you have any further constructive comments, we would be delighted to receive them.
Thank you.
Regards,
Nur Bahijah Mustapa, MSc. Students,
Faculty of Mechanical Engineering & Technology,
University Malaysia Perlis (UniMAP).